# Awareness and Related Factors of Dyslipidemia in Menopausal Women in Korea

**DOI:** 10.3390/healthcare10010112

**Published:** 2022-01-06

**Authors:** Jeonghee Jeong, Mijin Kim

**Affiliations:** 1Department of Nursing Science, Kyungsung University, Busan 48434, Korea; loveu1105@ks.ac.kr; 2Department of Nursing, Daegu Haany University, Gyeongsan-si 38610, Korea

**Keywords:** dyslipidemia, menopausal, women

## Abstract

This study aims to identify the awareness of dyslipidemia and the factors affecting it in menopausal women to prevent cardiovascular disease, a major cause of female mortality. This study used data from 2019, the first year of the eighth (2019–2021) Korea National Health and Nutrition Examination Survey conducted by the Korea Disease Control and Prevention Agency. A total of 975 women fulfilled the selection criteria. Dyslipidemia awareness and the related factors were analyzed with SPSS 26.0 complex sample software. Only 27.3% of menopausal women over age 40 with dyslipidemia were aware of the condition. Factors affecting their awareness level were age, subjective health awareness, body mass index, and underlying disease. The prevalence of dyslipidemia in menopausal women was high, but their awareness was significantly low. This finding confirms the need for measures to improve dyslipidemia awareness to prevent cardiovascular diseases in menopausal women.

## 1. Introduction

Cardiovascular disease is a major cause of death among women worldwide. Although there are several causes for cardiovascular disease, an important factor that directly affects its onset and progression is dyslipidemia [1,2].

After menopause, women experience serum lipid changes owing to a significant increase in the sex hormone estrogen [3]. Their low-density lipoprotein cholesterol (LDL-C), total cholesterol (TC), and triglycerides (TG) increase and high-density lipoprotein cholesterol (HDL-C) decreases [4]. In Korea, dyslipidemia among women increased with age and showed a significant difference before and after menopause. The prevalence was 27.6% in women aged ≤40 years, 55.9% in women aged 40–59 years, and 64.6% in women aged ≥60 years [5]. In particular, the prevalence of LDL-C was more than six times higher in those in their 50s when compared with those in their 30s [6]. Menopausal women develop sexual dysfunction due to vascular problems in the pelvis. They experience changes in serum lipid levels, and there is a significant increase in the incidence of fatal cardiovascular disease [3,7,8]. Dyslipidemia is associated with atherosclerosis in the process of causing cardiovascular disease. Cholesterol accumulated due to dyslipidemia is oxidized and increases the expression of intercellular adhesion molecule (ICAM)-1 and endothelial-selectin (E-selectin) for monocyte adhesion, thereby resulting in monocyte influx and cytokine production. The monocytes differentiate into macrophages and secrete monocyte chemoattractant protein (MCP)-1 to further promote the influx of monocytes. In addition, monocytes secrete cytokines, such as interleukin (IL)-6, and enhance the oxidation of cholesterol through the release of oxidizing substances. Macrophages absorb oxidized cholesterol and become foam cells, which are deposited on the walls of the blood vessels. This process results in the formation of plaque and causes atherosclerosis. In this manner, dyslipidemia increases the risk of atherosclerosis and cardiovascular disease [9,10,11,12,13,14]. Women with HDL-C levels < 50 mg/dL have a 30% increased risk of death from cardiovascular disease, and those with a TC level between 200 and 399 mg/dL have a 65% increased risk of death [15].

We should be mindful of the transition of menopausal women’s health from a low-health risk category to a high-health risk category when compared with their premenopausal health [16]. As dyslipidemia usually has no clear symptoms, it is difficult to make an early diagnosis. Even if it is confirmed through a health checkup, the level of awareness about the risk of disease is low. Hence, it is often left unattended, and people do not visit the hospital to receive an accurate diagnosis and treatment [17]. Disease awareness affects health behavior and change behavior for health management. It improves disease management attitude and treatment adherence and reduces disease prevalence and comorbidities [18]. When compared with chronic diseases such as diabetes and hypertension, dyslipidemia awareness is significantly lower in many countries, including Korea. The treatment rate is also low due to the low awareness level [19,20]. However, recognizing the risk of dyslipidemia is an important factor in cardiovascular disease prevention in menopausal women.

Most studies on dyslipidemia involve adults, and research with menopausal women has mostly been done to confirm that menopause is a risk factor for cardiovascular disease [21]. Investigations have also been performed on the prevalence and related factors of dyslipidemia in menopausal women [16] as well as on hormonal therapy for estrogen deficiency after menopause [15]. However, considering that awareness of dyslipidemia is significantly low, studies that confirm this awareness in menopausal women and identify the factors affecting it are greatly needed. Therefore, this study seeks to identify dyslipidemia awareness in women after menopause and the factors affecting it.

## 2. Methods

### 2.1. Research Design

This secondary data analysis studied the correlations and explored dyslipidemia awareness in menopausal women and the factors affecting it.

### 2.2. Research Data and Subjects

This study used data from 2019, the first year of the eighth Korea National Health and Nutrition Examination Survey (KNHANES) (2019–2021) conducted by the Korea Disease Control and Prevention Agency. KNHANES is a nationally conducted survey that aims to understand the health and nutritional status of people. The sampling frame was the 2010 Population and Housing Census (approximately 300,000 enumeration blocks), and a two-step stratified cluster sampling used the enumeration blocks and households as the primary and secondary sampling units, respectively. In the case of the first year of the 8th KNHANES (2019), the sampling frame was stratified based on city/province, dong/eup/myeon, and housing type (general housing, apartment). The residential area ratio, age of the head of the household, the ratio of single-person households, etc., were used as implicit stratification criteria to extract representative samples of citizens aged one year or older residing in Korea. In the first year of the 8th KNHANES (2019), there were 10,859 survey subjects and 8110 participants, and the participation rate was 74.7% [22].

The selection criteria for this study’s subjects were women over 40 years of age who responded with “natural menopause” or “artificial menopause” to the question on menstruation status, women having dyslipidemia, and those who responded to dyslipidemia awareness-related variables. Individuals who met the above four criteria were included. Individuals were categorized as having dyslipidemia if they satisfied one or more of the following [23]: (1) TC of ≥200 mg/dL, (2) HDL-C of <40 mg/dL, (3) LDL-C of ≥130 mg/dL, or (4) TG of ≥150 mg/dL or more. A total of 975 subjects satisfied all the above selection criteria (Figure 1).

### 2.3. Variable Selection

#### 2.3.1. General Characteristics

Age, area of residence, education level, job status, household income level, spouse status, and menopause period were the seven items selected. Age was classified as 40s, 50s, 60s, and 70s or older; the residential area was classified as urban (dong) and rural (eup/myeon); and education level was classified as elementary school or less, middle school, high school, and college or higher. Occupational status was classified as “yes” if the individual was currently economically active and “no” if economically inactive. Household income was classified into quartiles according to the average monthly household equalized income. According to the quartile criteria for household income level presented in the 8th KNHANES 1st year raw data utilization guidelines [22], an income of <1.06 million won was classified as low, between 1.06 million and 2.02 million won as lower-middle, between 2.02 million and 3.17 million won as upper-middle, and >3.17 million won as high. Spouse status was classified as “have a spouse” for individuals living with their spouse and “no spouse” for unmarried, separated, widowed, or divorced individuals. The menopause period was classified as <10 years (early postmenopause) and >10 years (late postmenopause) [24].

#### 2.3.2. Health-Related Characteristics

These characteristics included perceived health status, health screening, smoking, drinking amount, regular exercise, diet, body mass index (BMI), underlying diseases (hypertension and diabetes mellitus), and family history (hypertension, diabetes mellitus, and hyperlipidemia). Perceived health status was classified as “bad,” “normal,” and “good.” Health screening status was divided into “yes” and “no” based on whether the individual had a health checkup in the past 2 years. Smoking status was classified as “current smoker” for people who have smoked ≥5 packs of cigarettes in their entire life and currently or occasionally smoked, “ex-smoker” for people who have smoked ≥5 packs in their lifetime but are not currently smoking, and “non-smoker” for people who smoked <5 packs in their lifetime or never smoked. For people with experience drinking alcohol and who have drunk more than once a month in the past year, drinking was classified as “≤2 glasses,” “3–6 glasses,” or “≥7 glasses,” regardless of the type of alcohol. Those who had no experience drinking or those with experience drinking but drank less than once a month in the past year or did not drink were classified as “do not drink.” Regular exercise was classified as “yes” and “no” depending on whether moderate-intensity physical activity was performed for at least 150 min a week or high-intensity physical activity for at least 75 min a week [25]. Diet was divided into “yes” and “no” depending on whether a diet was followed for disease or weight control reasons. BMI was calculated using height and weight data by applying the formula weight (kg)/height (m^2^) and was classified as “normal” when the BMI was <22.9 kg/m^2^, “overweight” when the BMI was between 22 and 25.9 kg/m^2^, and “obese” when the BMI was >25 kg/m^2^, according to the World Health Organization Asia–Pacific region and the obesity diagnosis criteria of the Korean Society for Obesity [26,27]. Underlying diseases were classified as “none,” “hypertension only,” “diabetes mellitus only,” or “both,” depending on whether they had a doctor’s diagnosis of hypertension and diabetes mellitus. Family history of hypertension, diabetes mellitus, and hyperlipidemia was divided into “yes” and “no.”

#### 2.3.3. Dyslipidemia Awareness Level

The level of dyslipidemia awareness was defined when the individuals responded with a “yes” to having a doctor’s diagnosis of dyslipidemia or “yes” to currently having the disease or in treatment or medication.

### 2.4. Data Analysis Method

The KNHANES was assessed using the complex sample design method. To enhance the estimation accuracy, a complex sample analysis that considered strata, clusters, and weight was performed using the SPSS 26.0 software. Statistical significance was set at 0.05. The subjects’ general characteristics, health-related characteristics, and dyslipidemia awareness were calculated as unweighted frequency and weighted percentages for categorical data and mean and standard error for continuous data. The Rao–Scott chi-square test was performed to analyze the dyslipidemia awareness level according to the participants’ general and health-related characteristics. For dyslipidemia awareness according to the subjects’ general and health-related characteristics, the odds ratio and 95% confidence interval were obtained through simple logistic regression analysis. Multiple logistic regression analysis was performed by using the significant variable in the simple logistic regression analysis as an explanatory variable and dyslipidemia recognition as an outcome variable for the factors related to the subjects’ awareness of dyslipidemia. Odds ratio and 95% confidence interval were obtained from the analysis.

### 2.5. Ethical Considerations

This study was conducted after the Kyungsung University Institutional Review Board exempted it from review (KSU-21-09-001).

## 3. Results

### 3.1. Subjects’ General and Health-Related Characteristics and Awareness Level on Dyslipidemia

Table 1 presents the general characteristics of the subjects and their awareness on dyslipidemia. Among menopausal women over 40 years of age who had dyslipidemia, only 27.3% were aware of it.

Most subjects were in their 50s (39.9%) and 60s (32.2%). Dyslipidemia awareness was high among the subjects in their 60s (33.6%) and ≥70s (32.2%). Most of the subjects were in the “elementary school or less” category in terms of educational level (34.8%), and dyslipidemia awareness was also the highest at 34.4% in the “elementary school graduate or lower” group. As for the household income level, the percentage of subjects with “low” income was the highest (28.4%), and dyslipidemia awareness was high in the “lower-middle” (27.9%) and “low” (32.2%) income groups. The majority responded that they “have a spouse” (69.8%), and dyslipidemia awareness was high in the “no spouse” group (30.2%). The percentage of people with a menopause period “≥10 years” was 54.5%, and dyslipidemia awareness was also high at 34.0% in this group (Table 1).

Table 2 shows the subjects’ health-related characteristics and dyslipidemia awareness level. Most subjects perceived their health status as “normal” (54.9%), and dyslipidemia awareness was the highest in the “bad” group (40.5%). As for drinking, 70.5% of the respondents said that they “do not drink,” and dyslipidemia awareness was also high at 30.3% in such people. As for BMI, 38.5% of the subjects were “normal,” and 35.2% were “obese.” Dyslipidemia awareness was 34.3% in the “obese” group. As for the underlying disease, 24.3% had “hypertension only”; 3.2% had “diabetes mellitus only”; 7.9% had “both”; and 64.5% of the respondents had no underlying diseases. Dyslipidemia awareness was the highest in participants with “diabetes mellitus only” (60.7%) and “both” (66.2%) (Table 2).

### 3.2. Factors Related to the Subjects’ Dyslipidemia Awareness Level

Table 3 shows the results obtained from the multiple logistic regression analysis, following the presence or absence of dyslipidemia awareness, to identify the factors affecting dyslipidemia awareness level. The results showed that these factors were age, perceived health status, BMI, and presence of underlying diseases.

As observed, the dyslipidemia awareness level was 12.77 times higher in those in their 50s than in those in their 40s and 14.71 times higher in those in their 60s than in those in their 40s. With regard to the perceived health status, the awareness was 2.10 times higher when it was “bad” than when it was “good.” Additionally, higher dyslipidemia awareness was observed when the BMI was “obese,” and the levels were 1.51 times higher than when the BMI was “normal.” For the presence of underlying diseases, compared with “none,” the awareness was 2.57 times higher when the subjects had “hypertension alone”; 6.12 times higher when they had “diabetes only”; and 7.60 times higher when they had “both.”

While education level, household income level, marital status, menopause period, and drinking quantities also showed significant differences during simple logistic regression analysis, no significance was observed in multiple regression analysis results (Table 3).

## 4. Discussion

In this research, by ascertaining that the awareness level on dyslipidemia in menopausal women was significantly low and by establishing the factors affecting this awareness, an urgent need to improve the awareness level after menopause has been confirmed.

The prevalence of dyslipidemia among menopausal women over 40 years of age was 59.3% (among 1645 menopausal women over 40 years of age, 975 of them had dyslipidemia. See Figure 1), whereas only 27.3% of them were aware of the condition known as “dyslipidemia.” As observed, factors affecting dyslipidemia awareness level were age, perceived health status, BMI, and presence of underlying diseases.

The results of a previous study showed that the prevalence of dyslipidemia before menopause was 35.0%, which increased to as high as 65.2% after menopause [4]. Additionally, in a study of adults who were ≥19 years of age and did not have a history of cardiovascular disease, the prevalence of dyslipidemia was 15.5% and the awareness level was 76.9%. However, the awareness level in this study was only 27.3%, which was quite low [28]. This value was lower than that of the awareness level among adults over 18 years of age (31.0%) [29]. Therefore, although the period of menopause is proposed to be related to the characteristics in menopausal women who managed the menopausal symptoms through self-regulation, the women did not perform well with professional health management even though they experienced abnormal menopause-related symptoms [30]. In short, due to the lack of an accurate awareness on the clinical symptoms of dyslipidemia and the risk of the disease, the women are unaware of the need for professionally managing early diagnosis and early treatment, which is predicted to have failed to sustain the healthcare behavior. Hence, for proper dyslipidemia management of menopausal women, an accurate awareness of the disease is required. Such an understanding would contribute to the lowering of the risk for cardiovascular disease.

Data obtained from these results also suggested that menopausal women had higher risks of cardiovascular diseases due to the lower treatment rate observed during the low dyslipidemia awareness study period.

Among menopausal women, the awareness level of those in their 50s was 12.77 times higher than those in their 40s, whereas those in their 60s had 14.71 times higher awareness. The high dyslipidemia awareness among people in their 50s and 60s could be attributed to the fact that they were more sensitive to the risk of exposure to chronic diseases than those in their 40s. Therefore, they paid more attention to their health and abnormal menopause-related symptoms. In this study, the cognitive level of dyslipidemia among people >70 years of age was higher than that of those in their 50s and 60s, but the prevalence decreased slightly as in previous studies [5]. This decrease does not appear to be a direct influencing factor because the overall level of awareness and management of chronic diseases, such as hypertension and diabetes, which can occur with increasing age is high. However, this study showed that those in their 40s should also pay attention to these symptoms. Furthermore, although the timing of the onset of menopause varies slightly from country to country, it mainly occurs between the ages of 49 and 52 [31]. Hence, when compared with those who experienced menopause in their 50s or later than that, the risk of cardiovascular disease was 1.55 times higher in women who experienced menopause before 40 years of age, 1.30 times higher in those who experienced it within 40–45 years of age, and 1.12 times higher in those who experienced it within 45–49 years of age [32]. As such, increased risks of cardiovascular disease in women in their 40s, in addition to the comparatively low awareness level of dyslipidemia in the 40s, is of great significance. Dyslipidemia tests in Korea (TC, LDL-C, TG, and HDL-C) are provided for those over 40 years of age. However, considering the high prevalence and low awareness on dyslipidemia among the menopausal women, the method of screening and continuous monitoring for dyslipidemia right from those in their 20s should be considered, as suggested by the American Heart Association [33]. The health checkup was not meaningful in this study, and it is necessary to confirm its effectiveness through continuous research in the future. Just as education on physical changes in adolescents is given right before entering adolescence, education on physical changes experienced during menopause should be provided beforehand.

The subjective perceived health status and BMI were confirmed to be the predictors of dyslipidemia awareness [20]. In this study, dyslipidemia awareness level was twice higher when the subjects perceived their health status to be bad. This perception was proposed to be because of activities to acquire more pathological information, in addition to more frequent visits to medical services for health management [34], as people indulged in active health management activities when they perceived their health status as bad [20]. Conversely, if people do not recognize that their health is bad, they will engage in relatively less active health management activities. Therefore, it is necessary to identify the cause as to why people perceive their health to be good even when they have dyslipidemia. Further studies should also explore the ways by which people can improve the awareness and manage their health by themselves.

An international study on obesity discovered that more than 39% of women experiencing menopause were either overweight or obese [35]. This status was due to weight gain because of changes in hormones and body fat distribution, in addition to unhealthy lifestyles [36]. In this study, overweight and obese subjects accounted for 61.5% of those with dyslipidemia, which was much higher than the international study results. This finding could be related to the relatively low weight management behavior, unlike the high level of awareness on obesity in Korean middle-aged women [37]. Also, for people whose BMI results fell in the obesity category, their dyslipidemia awareness was 1.51 times higher. This result might be because obese people perceived themselves to be at a high risk for developing various chronic diseases, such as obesity, which is one of the direct risk factors for high blood pressure, diabetes, and dyslipidemia [38]. Nevertheless, menopausal women with normal BMI should be studied. People think that obesity is associated with lipid-related problems. Therefore, it is proposed that if the body weight is normal, people do not recognize themselves to be at risk for any disease even if they were afflicted with dyslipidemia. Preventive and management educational steps to inform people that dyslipidemia can occur even in those with a normal weight, including ways to recognize those that are also exposed to the risks of developing cardiovascular disease, is necessary. Therefore, obesity management is essential to prevent cardiovascular disease in overweight and obese people.

The awareness level on dyslipidemia among menopausal women was 2.57 times higher when they had hypertension as an underlying disease, 6.12 times higher when they had diabetes, and 7.60 times higher when they had both hypertension and diabetes. Dyslipidemia is therefore highly correlated with hypertension and diabetes, thereby acting simultaneously as a cause for cardiovascular diseases [7,39,40]. In addition, the formation of adipose tissue in obese people is a factor that increases vascular inflammation, oxidative stress, and atherosclerosis. Furthermore, the risk increases significantly when diabetes and hypertension are also present [41,42]. Hence, considering that 61.5% were either overweight or obese in this study, people with high blood pressure or diabetes would have been provided with information, education, and treatment for the prevention and management of dyslipidemia during the course of treating the underlying diseases. Their cognitive level of dyslipidemia was therefore likely to be high. However, women without underlying diseases have lower awareness levels because they believe that they are healthy. This may be the reason why low dyslipidemia awareness levels were observed among individuals who perceived themselves as having good health.

While this research has a representative characteristic because it used data from KNHANES, it is a cross-sectional study that has a limitation in its analysis of causality. Furthermore, because the study only focused on general and health-related characteristics during the analysis of dyslipidemia-related factors in menopausal women, it is also limited in generalization. Hence, studies that consider the start time of menopause, the period of menopause, the time of dyslipidemia diagnosis, and the duration of the illness in postmenopausal women are necessary in the future.

## 5. Conclusions

In contrast to the high prevalence of dyslipidemia among menopausal women, it has been confirmed that the awareness level about the disease is significantly low in this group. Therefore, to reduce the risks of cardiovascular disease and mortality rate, early detection of dyslipidemia in menopausal and young women who have not experienced menopause is required. Moreover, measures should be taken to increase the awareness level of those at risk for the disease and make them understand the importance of early detection and treatment.

## Figures and Tables

**Figure 1 healthcare-10-00112-f001:**
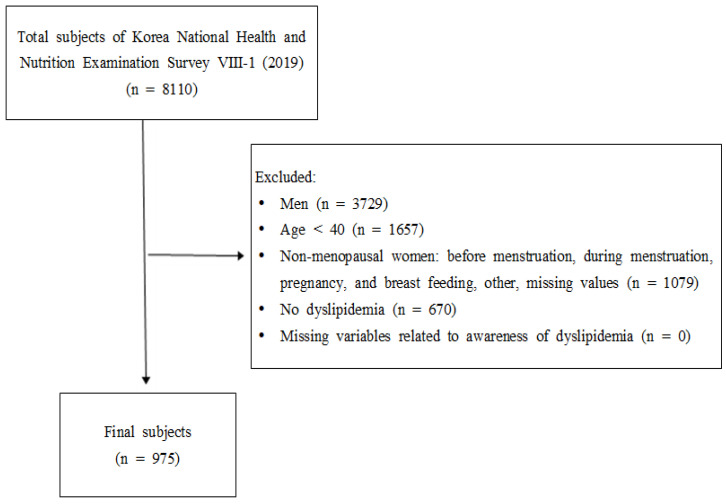
Flowchart depicting the selection of study participants.

**Table 1 healthcare-10-00112-t001:** Demographic characteristics and dyslipidemia awareness level in menopausal women (*n* = 975).

Characteristics		*n* (%)	Dyslipidemia Awareness			*p*
			Rate	Yes, *n* (%)	No, *n* (%)	
Total			27.3	270 (100)	705 (100)	
Age (yrs)	40–49	25 (3.0)	2.1	1 (0.2)	24 (4.1)	<0.001
	50–59	339 (39.9)	21.0	72 (30.7)	267 (43.3)	
	60–69	335 (32.2)	33.6	111 (39.7)	224 (29.4)	
	≥70	276 (24.6)	32.2	86 (29.4)	190 (23.2)	
	M ± SE	62.85 ± 0.38				
Residence	County	233 (21.0)	25.8	65 (19.9)	168 (21.4)	0.611
	City	742 (79.0)	27.6	205 (80.1)	537 (78.6)	
Educational level * (*n* = 965)	Elementary school or less	367 (34.8)	34.4	125 (43.8)	251 (31.4)	0.004
	Middle school	148 (14.7)	27.7	45 (14.9)	103 (14.6)	
	High school	289 (33.1)	24.2	69 (29.3)	220 (34.5)	
	College or higher	161 (17.4)	18.7	31 (12.0)	130 (19.5)	
Currently working	No	541 (54.9)	30.0	165 (60.4)	376 (58.2)	0.059
	Yes	434 (45.1)	23.9	105 (39.6)	329 (47.2)	
Household Income (in quartile) * (*n* = 970)	High	224 (24.3)	20.1	48 (18.1)	176 (26.6)	0.062
	Upper-middle	215 (23.8)	27.1	59 (23.9)	156 (23.8)	
	Lower-middle	238 (23.5)	27.9	70 (24.2)	168 (23.2)	
	Low	293 (28.4)	32.2	91 (33.7)	202 (26.4)	
Spouse	Yes	659 (69.8)	24.9	172 (63.7)	487 (72.1)	0.031
	No	316 (30.2)	32.8	98 (36.3)	218 (27.9)	
Menopause period (yrs) * (*n* = 965)	<10	393 (45.5)	19.2	79 (32.1)	314 (50.6)	<0.001
	≥10	572 (54.5)	34.0	189 (67.9)	383 (49.4)	
	M ± SE	12.91 ± 0.43				

M, mean; SE, standard error; yrs, years. Data are expressed as weighted percentages. * Different from total number (*n* = 975) due to missing values.

**Table 2 healthcare-10-00112-t002:** Health-related characteristics and dyslipidemia awareness level in menopausal women (*n* = 975).

Characteristics		*n* (%)	Dyslipidemia Awareness			*p*
			Rate	Yes, *n* (%)	No, *n* (%)	
Total			27.3	270 (100)	705 (100)	
Perceived health status	Good	226 (22.9)	20.3	44 (17.0)	182 (25.1)	<0.001
	Not bad	518 (54.9)	24.8	131 (50.0)	387 (56.8)	
	Bad	231 (22.2)	40.5	95 (33.0)	136 (18.2)	
Health screening * (*n* = 974)	No	248 (26.1)	32.0	72 (30.6)	176 (24.4)	0.104
	Yes	726 (73.9)	25.6	198 (69.4)	528 (75.6)	
Smoking * (*n* = 973)	Non-smoker	890 (90.9)	26.4	240 (87.9)	650 (92.0)	0.203
	Ex-smoker	41 (4.0)	33.4	14 (4.9)	27 (3.6)	
	Current smoker	42 (5.1)	38.6	16 (7.2)	26 (4.3)	
Drinking amount * (*n* = 973)	None	701 (70.5)	30.3	210 (78.1)	491 (67.7)	0.045
	≤2 glass	148 (15.3)	19.4	32 (10.9)	116 (16.9)	
	3–6 glass	97 (11.3)	20.9	22 (8.7)	75 (12.3)	
	≥7 glass	27 (2.9)	22.6	6 (2.4)	21 (3.1)	
Exercise * (*n* = 974)	No	886 (90.9)	27.2	248 (90.7)	638 (91.0)	0.890
	Yes	88 (9.1)	28.0	22 (9.3)	66 (9.0)	
Diet * (*n* = 883)	No	618 (69.7)	25.6	159 (65.8)	459 (71.1)	0.182
	Yes	265 (30.3)	30.6	86 (34.2)	179 (28.9)	
Body mass index (kg/m^2^) * (*n* = 971)	<22.9 (normal)	374 (38.5)	20.5	81 (28.9)	293 (42.0)	0.003
	23.0–24.9 (overweight)	250 (26.3)	27.8	66 (26.8)	184 (26.1)	
	≥25 (obese)	347 (35.2)	34.3	122 (44.3)	225 (31.8)	
Underlying disease	None	621 (64.5)	17.4	112 (41.2)	509 (73.3)	<0.001
	Hypertension only	241 (24.3)	36.2	89 (32.3)	152 (21.3)	
	Diabetes mellitus only	35 (3.2)	60.7	18 (7.2)	17 (1.8)	
	Both	78 (7.9)	66.2	51 (19.3)	27 (3.7)	
Family history * (*n* = 892)	No	505 (55.0)	26.1	131 (54.4)	374 (55.2)	0.859
	Yes	387 (45.0)	26.7	111 (45.6)	276 (44.8)	

Data are expressed as weighted percentages. * Different from total number (*n* = 975) due to missing values.

**Table 3 healthcare-10-00112-t003:** Factors associated with dyslipidemia awareness in menopausal women (*n* = 975).

Characteristics		Simple Logistic Regression		Multiple Logistic Regression	
		Adjusted OR (95% CI)	*p*	Adjusted OR (95% CI)	*p*
Age (yrs)	40–49	1		1	
	50–59	12.32 (1.56–97.32)	0.018	12.77 (1.59–102.58)	0.017
	60–69	23.44 (3.10–177.06)	0.002	14.71 (1.82–118.74)	0.012
	≥70	21.98 (2.78–173.50)	0.004	7.50 (0.82–68.29)	0.073
Residence	County	1			
	City	1.10 (0.77–1.56)	0.611		
Educational level	Elementary school or less	1		1	
	Middle school	0.73 (0.44–1.21)	0.218	1.20 (0.67–2.15)	0.534
	High school	0.61 (0.44–0.85)	0.004	1.13 (0.66–1.93)	0.663
	College or higher	0.44 (0.28–0.69)	<0.001	1.12 (0.58–2.15)	0.742
Currently working	No	1			
	Yes	0.73 (0.53–1.01)	0.059		
Household income (in quartile)	High	1		1	
	Upper-middle	1.48 (0.90–2.45)	0.124	1.23 (0.74–2.04)	0.424
	Lower-middle	1.54 (0.93–2.54)	0.091	0.99 (0.57–1.72)	0.976
	Low	1.89 (1.21–2.93)	0.005	1.07 (0.58–1.96)	0.825
Spouse	Yes	1		1	
	No	1.47 (1.04–2.10)	0.032	1.27 (0.83–1.97)	0.273
Menopause period (yrs)	<10	1		1	
	≥10	2.17 (1.56–3.01)	<0.001	1.57 (0.98–2.51)	0.058
Perceived health status	Good	1		1	
	Not bad	1.30 (0.83–2.03)	0.255	1.31 (0.80–2.13)	0.284
	Bad	2.68 (1.70–4.22)	<0.001	2.10 (1.27–3.48)	0.004
Health Screening	No	1			
	Yes	0.73 (0.50–1.07)	0.105		
Smoking	Non-smoker	1			
	Ex-smoker	1.40 (0.67–2.91)	0.371		
	Current smoker	1.75 (0.84–3.62)	0.133		
Drinking amount	None	1		1	
	≤2 glass	0.56 (0.33–0.94)	0.027	0.67 (0.39–1.15)	0.147
	3–6 glass	0.61 (0.35–1.08)	0.088	0.79 (0.42–1.49)	0.466
	≥7 glass	0.67 (0.27–1.70)	0.398	0.63 (0.20–1.97)	0.420
Exercise	No	1			
	Yes	0.89 (0.58–1.86)	0.890		
Diet	No	1			
	Yes	1.28 (0.89–1.84)	0.182		
Body mass index (kg/m^2^)	<22.9 (normal)	1		1	
	23.0–24.9 (overweight)	1.49 (0.96–2.32)	0.076	1.22 (0.75–1.96)	0.424
	≥25 (obesity)	2.02 (1.41–2.91)	<0.001	1.51 (1.04–2.20)	0.033
Underlying disease	None	1		1	
	Hypertension only	2.70 (1.92–3.79)	<0.001	2.57 (1.67–3.98)	<0.001
	Diabetes mellitus only	7.35 (3.17–17.01)	<0.001	6.12 (2.58–14.51)	<0.001
	Both	9.31 (5.21–16.63)	<0.001	7.60 (3.99–14.47)	<0.001
Family history	No	1			
	Yes	1.03 (0.73–1.47)	0.859		

CI, confidence interval; OR, odds ratio

## Data Availability

Not applicable.

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
