# Peer review of "Awareness and Related Factors of Dyslipidemia in Menopausal Women in Korea"

_healthcare, 2022, doi:10.3390/healthcare10010112_

Round 1
Reviewer 1 Report
Please see the attachment.

Reviewer 2 Report
In this report, Jeong&Kim analyze the underlying factors influencing the awarness of dyslipidemia among Korean females. Their study was well-conducted and the paper is also well structured and clear.
They find that age, perceived health status, BMI, and presence of underlying diseases, are the main factors affecting dyslipidemia's awareness level. However, the authors are concerned by the fact that dyslipidemia is still present in a great proportion of menopausal women that have not any awarness about their lipid profile.
This research could be useful to carry-out preventive public health strategies among Korean population.
Just, please arrange the p-value columns into the tables 1 and 2.
Author Response
Thank you for your review, which have helped us improve the quality of this manuscript. According to the reviewers’ comments, we have made revision on the manuscript. We used violet colored fonts to indicate the revised parts for your recognition. If the reviewer's comments overlap, it's indicated in red.
Comments : please arrange the p-value columns into the tables 1 and 2.
Response : Arrange the p-value columns into the tables 1 and 2 was edited.

Reviewer 3 Report
The manuscript is well written and interesting. I would suggest to underline how dyslipidemia also represents an important component of the metabolic syndrome and how it increases the risk of severe cardiovascular diseases and female sexual dysfunction (an early indicator of vascular damage), especially in menopause, through chronic vascular inflammation, oxidative stress and atherosclerosis. (bibliography:
- Di Francesco S., Caruso M., Robuffo I., Militello A., Toniato E. The Impact of Metabolic Syndrome and its Components on Female Sexual Dysfunction: A Narrative Mini-Review. Curr Urol 2019;12:57-63.;
- Servadei F, Anemona L, Cardellini M, Scimeca M, Montanaro M, Rovella V, Di Daniele F, Giacobbi E, Legramante IM, Noce A, Bonfiglio R, Borboni P, Di Daniele N, Ippoliti A, Federici M, Mauriello A. The risk of carotid plaque instability in patients with metabolic syndrome is higher in women with hypertriglyceridemia. Cardiovasc Diabetol. 2021 May 6;20(1):98.)
I also recommend to review table 2 in particular to better format the value of p.
Author Response
Thank you for your review, which have helped us improve the quality of this manuscript. According to the reviewers’ comments, we have made revision on the manuscript. We used green colored fonts to indicate the revised parts for your recognition. If the reviewer's comments overlap, it's indicated in red.
Comments
- I would suggest to underline how dyslipidemia also represents an important component of the metabolic syndrome and how it increases the risk of severe cardiovascular diseases and female sexual dysfunction (an early indicator of vascular damage), especially in menopause, through chronic vascular inflammation, oxidative stress and atherosclerosis.
- I also recommend to review table 2 in particular to better format the value of p.
Response
- It added information on the process of dyslipidemia that causes cardiovascular disease and female sexual dysfunction (line 26-27; 33-47).
- Arrange the p-value columns into the tables 1 and 2 was edited.

Reviewer 4 Report
This is an interesting report on the factors that contribute to whether or not dyslipidemia is recognized in postmenopausal women, but there are some issues that need to be resolved.
#1 There were a number of factors that are listed as contributing to the perception of dyslipidemia, but they seem to be insufficiently considered. As for age, the significant difference disappears for those over 70 years old, but there are probably more complications with hypertension and diabetes. I think that people with a higher BMI have more complications of diabetes and hypertension, but the multivariate analysis still shows a significant difference. This has also been discussed, but please discuss it in more detail. In the conclusion, you mention measurement as a solution, but health screening does not show a significant difference. A discussion of this may be necessary.
#2 In Table 1, the percentages seem not to be correct. For example, under residence, it says county 233 (21.0%), city 742 (79.0%), but I think 233 (23.9%), 742 (76.1%) is correct. Similarly, what are the percentages of awareness yes and no? I can't get these values when I calculate them. I think the percentages are displayed differently in each case. Also, the numbers were displayed to the second decimal place, but this was not clinically meaningful and should be displayed as an appropriate minority or integer.
#3 In Table 2, there are many items that do not total 975. For example, the total for diet is 883.
#4 In Figure 1, if n = 0, I think that item should be deleted.
Author Response
Thank you for your review, which have helped us improve the quality of this manuscript. According to the reviewers’ comments, we have made revision on the manuscript. We used blue colored fonts to indicate the revised parts for your recognition. If the reviewer's comments overlap, it's indicated in red.
Comments
- There were a number of factors that are listed as contributing to the perception of dyslipidemia, but they seem to be insufficiently considered. As for age, the significant difference disappears for those over 70 years old, but there are probably more complications with hypertension and diabetes. I think that people with a higher BMI have more complications of diabetes and hypertension, but the multivariate analysis still shows a significant difference. This has also been discussed, but please discuss it in more detail. In the conclusion, you mention measurement as a solution, but health screening does not show a significant difference. A discussion of this may be necessary.
- In Table 1, the percentages seem not to be correct. For example, under residence, it says county 233 (21.0%), city 742 (79.0%), but I think 233 (23.9%), 742 (76.1%) is correct.
- What are the percentages of awareness yes and no? I can't get these values when I calculate them. I think the percentages are displayed differently in each case.
- The numbers were displayed to the second decimal place, but this was not clinically meaningful and should be displayed as an appropriate minority or integer.
- In Table 2, there are many items that do not total 975. For example, the total for diet is 883.
Response
- The reason why there was no significant difference in the 70s was further described (line 255-260; 274-275; 312-319). The relationship between hypertension and diabetes and BMI was described. (line 312-319). The contents of the health examination were modified. (line 274-275).
- As described in Data Analysis Method, weighted percentages and unweighted frequencies are presented in the table because complex sample analysis applies weights to the analysis. This can lead to differences in percentages.
- In the table 1 & 2, Rate is the weighted row percentage, and is the rate of people who recognized dyslipidemia. On the other hand, Yes (%) and No (%) represent weighted column percentages.
- Usually, the mean, standard error, OR value, and Cl value are indicated to the second decimal places, so this was followed.
- The difference in frequency by variable is due to missing values.
Missing values ​​of non-applicable and non-response existing in individual variables were designated as discrete missing values. In the case of complex sample analysis, missing values ​​should be treated as valid values, so data were analyzed after ‘treating them as valid values’ in the analysis options.
The content of missing values ​​was added to the data analysis method (line 153-155).

Round 2
Reviewer 1 Report
Thank you for the reply. But there are a few more things that need to be revised.
* Reserch data and subject
- In complex sampling analysis, even if a missing value is designated as a 'valid value', it is excluded from the analysis. The reason that missing values should be designated as 'valid values' is because of the complex sample design. Therefore, only variables related to awareness would not be excluded from the analysis as missing values.
*Result
1. Looking at Table 1 and Table 2, the number of subjects for some variables is not 975. For example, the education level, household income, and menopause period in table1 are not 975. Also in Table 2, the number of subjects for most variables is not 975.
Author Response
Thank you for your review, which have helped us improve the quality of this manuscript. According to the reviewers’ comments, we have made revision on the manuscript. We used orange colored fonts to indicate the revised parts for your recognition.
Comments:
- In complex sampling analysis, even if a missing value is designated as a 'valid value', it is excluded from the analysis. The reason that missing values should be designated as 'valid values' is because of the complex sample design. Therefore, only variables related to awareness would not be excluded from the analysis as missing values.
- Looking at Table 1 and Table 2, the number of subjects for some variables is not 975. For example, the education level, household income, and menopause period in table 1 are not 975. Also in Table 2, the number of subjects for most variables is not 975.
Response:
- There was a lack of understanding of the reviewer's comments. It is true that missing values ​​are excluded from data analysis even if missing values ​​are treated as valid values. There are variables that differ from total N in independent variables due to missing values, so they are shown in Tables 1 & 2. In the previous answers, the method of handling missing values ​​was deleted from the data analysis method.
- Differences in numbers due to missing values ​​are indicated (Tables 1 & 2).